# Subcritical Water Extraction to Valorize Grape Biomass—A Step Closer to Circular Economy

**DOI:** 10.3390/molecules28227538

**Published:** 2023-11-11

**Authors:** Cátia Ferreira, Manuela M. Moreira, Cristina Delerue-Matos, Mafalda Sarraguça

**Affiliations:** 1LAQV/REQUIMTE, Laboratório de Química Aplicada, Faculdade de Farmácia da Universidade do Porto, R. Jorge de Viterbo Ferreira 228, 4050-313 Porto, Portugal; catia2692000@gmail.com; 2LAQV/REQUIMTE, Instituto Superior de Engenharia do Porto, Instituto Politécnico do Porto, R. Dr. António Bernardino de Almeida 431, 4249-015 Porto, Portugal; manuela.moreira@graq.isep.ipp.pt (M.M.M.); cmm@isep.ipp.pt (C.D.-M.)

**Keywords:** biomass valorization, grape pomace, subcritical water extraction, circular economy, green chemistry, green extraction

## Abstract

With the increase in the world population, the overexploitation of the planet’s natural resources is becoming a worldwide concern. Changes in the way humankind thinks about production and consumption must be undertaken to protect our planet and our way of living. For this change to occur, sustainable development together with a circular economic approach and responsible consumption are key points. Agriculture activities are responsible for more than 10% of the greenhouse gas emissions; moreover, by 2050, it is expected that food production will increase by 60%. The valorization of food waste is therefore of high importance to decrease the environmental footprint of agricultural activities. Fruits and vegetables are wildly consumed worldwide, and grapes are one of the main producers of greenhouse gases. Grape biomass is rich in bioactive compounds that can be used for the food, pharmaceutical and cosmetic industries, and their extraction from this food residue has been the target of several studies. Among the extraction techniques used for the recovery of bioactive compounds from food waste, subcritical water extraction (SWE) has been the least explored. SWE has several advantages over other extraction techniques such as microwave and ultrasound extraction, allowing high yields with the use of only water as the solvent. Therefore, it can be considered a green extraction method following two of the principles of green chemistry: the use of less hazardous synthesis (principle number 3) and the use of safer solvents and auxiliaries (principle number 5). In addition, two of the green extraction principles for natural products are also followed: the use of alternative solvents or water (principle number 2) and the use of a reduced, robust, controlled and safe unit operation (principle number 5). This review is an overview of the extraction process using the SWE of grape biomass in a perspective of the circular economy through valorization of the bioactive compounds extracted. Future perspectives applied to the SWE are also discussed, as well as its ability to be a green extraction technique.

## 1. Introduction

In a space of one hundred years, the world’s population has increased more than four times, and in 2066, it is estimated that it will reach 10.43 billion people [1]. The abrupt population growth that has been observed in recent years derives from the improvement in the quality of life and in health and hygiene systems, which logically drastically reduces the number of deaths and is translated into population growth. However, the increase in the world population has several negative impacts on the planet, in which climate change, decreased food security, loss of biodiversity, destruction of ecosystems and overexploitation of the planet’s natural resources are the most visible ones [2]. The main cause of climate change is the emission of greenhouse gases, which retain solar radiation inside the planet, preventing its release into space and consequently contributing to global warming [3,4]. Between 2011 and 2020, the recorded global temperature of the Earth’s surface was 1.1 °C above the values registered between 1850 and 1900 [2,5]; in addition, the summer of 2022 was one of the hottest in Europe so far [5]. This increase in temperature is alarming, affecting the health and well-being of living beings and exposing the planet to a greater risk of catastrophic events. Global actions should be undertaken to be able to achieve a 55% reduction in greenhouse gas emissions by 2030 when compared with 1990 (Figure 1) [5]. Thus, to protect our planet, humankind must make changes in its way of life, ranging from quite simple changes in our daily lives to major changes at the industrial level. For this change, sustainable development, a circular economy, responsible consumption and the use of renewable energies are key points.

In this sense, the European Union has made efforts to reduce greenhouse gas emissions and, in 2015, the “Neutral climate now” plan was launched by the UNFCCC (United nations framework convention on climate change), which challenges all parts of the world to apply measures to achieve the same global goal: climate neutrality by 2050. The concept of climate neutrality implies that global greenhouse gas emissions will be reduced [3]. This must be a commitment made by us, as individuals, as society, by the governments and by all the industries. In addition, the European Environment Agency (EEA) demanded the exploration of new alternative paths that would allow the transition from a linear economy to a circular economy. A linear economy involves obtaining and processing raw materials that are then consumed in a short period of time and discarded as waste shortly after its use [6]. Alternatively, a circular economy encourages the recovery of waste by reusing, valuing, reintroducing and redistributing materials in the supply chain, which allows the lifespan of the products to be extended (Figure 2). The circular economy allows for economic growth based on the management of global resources, avoiding the production and use of novel resources and recycling the existing ones, generating in this way less waste, which consequently reduces the negative environmental impacts caused by them [6].

Agriculture and forestry (18%) and food applications (13%) were, between 2014 and 2020, some of the areas of the LIFE projects (the financial instrument for the environment) with the highest adherence toward achieving a circular economy (Figure 3).

The current food system (production, transport, processing, packaging, storage, retailing, consumption, losses and waste) feeds most the world’s population, and per-capita supply has increased by more than 30% since 1961 [8]. Despite its importance, agriculture is also a sector with a significant negative impact on the environment, due to the use of about half of the habitable land, to the use of substantial amounts of water, and perhaps the most important reason, to the high greenhouse gas emissions [9]. According to the EEA, 11% of the total greenhouse gas emissions in 2021 came from agriculture (Figure 4), which corresponds to around 378,430.47 kt of CO_2_ equivalents. By 2050, it is estimated that food production will increase by at least 60%, so greenhouse gas emissions will only tend to increase, as well as the production of waste and food by-products [6].

According to the FAO (Food and Agriculture Organization from the United Nations), the food supply chain could become one of the most greenhouse-gas-producing activities in the coming years due to the very rapid growth of food processing, packaging and transport, caused by the demand from a growing population [11]. Additionally, the increase in food production implies an increase in waste and by-products. Food waste is an enormous problem in ethical, social, economic and environmental terms. Food waste can be reused as a source of energy and transformed into organic fertilizer, animal feed, biopesticides and bioplastics or as a natural source of bioactive compounds [12,13]. There is an urgent need for the transition to a more sustainable food system with changes in food production, its processing, and waste management. Strategies have been developed around the world to reduce food waste and to achieve sustainable development.

The use and recovery of waste by-products are some of the measures that significantly reduce the emission of greenhouse gases into the environment, while at the same time reducing the volume of food waste in landfills [6,14]. Thus, researchers and industries worldwide are increasingly showing interest in the valorization of food waste and by-products contributing to a sustainable world and at the same time creating economic value [6,15,16].

## 2. Grape Biomass

Fruits and vegetables are the most consumed foods worldwide and, consequently, the largest producers of waste. Grapes are one of the main producers of greenhouse gases, as shown in Figure 5 [6], with 8.28 kg of CO_2_ equivalents emitted for each kilogram of grapes produced in 2022 [9].

Grapes are produced by plants of the genus *Vitis* consisting of about 60 species of vines, with *Vitis vinifera* L. being the most grown worldwide, with approximately 78 million tons produced every year [17,18,19,20,21,22]. There are several ways to consume grapes, namely fresh or processed in the form of wine, vinegar, juice, jam, jelly, dried grapes, oil or grape seed extract [20]. Grapes are one of the largest fruit crops in the world, and about 75% of its production is used for wine manufacture [17,20,23,24]. Wine manufacture extends to all countries worldwide and is currently embedded in their culture where it has great socioeconomic importance [20,25,26,27]. The largest wine-producing regions are in Europe (Italy, Spain, France, Germany and Portugal), America (the United States of America, Argentina and Chile), Australia and South Africa [21,28]. According to the International Organization of Vine and Wine (OIV), in 2022, the world wine production was around 259.9 mhl [29], and the top 10 wine-exporting countries were Italy, Spain, France, Chile, Australia, South Africa, Germany, Portugal, the USA and New Zeeland [29]. The top 10 wine-importers countries in 2022 were the USA, Germany, the UK, France, the Netherlands, Canada, Russia, China, Belgium and Portugal [29].

Wine production is one of the most economically important agricultural activities worldwide and, therefore, it is one of the sectors with the highest production of waste and by-products, which, when improperly managed, have potential negative impacts on the environment [21,30,31]. However, these residues and by-products are rich in bioactive and biodegradable compounds, so their proper management and revaluation are small steps toward sustainable development. Therefore, to promote a circular economy approach, the reduction in this food waste from the wine industry is encouraged. To this end, it is necessary that the management, revaluation and reuse of these residues constitute a priority.

Among the residues already mentioned, grape pomace is the main solid residue of the winemaking process resulting from the mechanical pressing of the whole grapes that generate the must [32,33]. By mass, grape pomace accounts for about 75% of the solid waste from the winemaking process and 20–30% of the total mass of grapes that are initially harvested [31,34,35]. When grape pomace accumulates in the soil, it constitutes a serious environmental problem, due to its antimicrobial and phytotoxic effects caused by the presence of antinutritive substances, and to the very low pH characteristic of the grape pomace [31].

In terms of composition, grape pomace is constituted by 25% of seeds and 25% of stalks, with the remaining 50% being grape skins [19,32,36]. Chemically, grape pomace consists of approximately 55–75% water, 30% polysaccharides and 6–15% proteins, lipids, sugars and unsaturated fatty acids [31]. In addition, it is a substrate rich in bioactive compounds, since during the winemaking process, about 70% of these compounds remain in the bagasse [31,37]. Thus, grape pomace represents a by-product of interest to the scientific community in the food, pharmaceutical and cosmetic industries, given its richness in bioactive compounds [21,27,30,37]. Its recovery and extraction are key points for the valorization of the by-products generated during wine processing.

### Bioactive Compounds of Grape Biomass

Positive effects on human health have been recognized from most products that are derived from grapes, due to the abundance of bioactive compounds that are present in their composition [38]. Bioactive compounds are secondary metabolites that are present in plants, such as fruits, vegetables and cereals, microorganisms and animals, and have the capacity to cause pharmacological or toxicological effects in humans and animals [39,40,41]. These compounds are classified into phenolic compounds (polyphenols), terpenoids and nitrogen-containing compounds (Figure 6). Phenolic compounds are the largest group of phytochemicals and are present in almost all plants in the form of secondary metabolites, where they play a key role both in growth and reproduction processes, as well as in the protection against pathogens and predators [42,43,44]. Regarding their chemical structure, polyphenols have one or more aromatic rings to which one or more hydroxyl groups are attached, so their structures can range from simple molecules, such as phenolic acids, to highly polymerized compounds, such as tannins [41,42,43,44]. Phenolic compounds can be further divided into flavonoids (anthocyanins, flavonols, flavones, isoflavones, flavanones and flavanols) and non-flavonoids (phenolic acids, stilbenes, lignins and tannins), as can be seen in Figure 6 [45].

The bioactive compounds most found in grape pomace are presented in Figure 7 and correspond to phenolic acids, stilbenes, flavonols, anthocyanins, lignins, flavonols and flavanols families [31,47,48,49,50,51]. Grape skin is mostly constituted of fibers, sugars, anthocyanins, flavonols (quercetin, myricetin and kaempferol) and tannins. Red grape skin may also contain crude protein, fat, ash, stilbenes (resveratrol), triterpenes and hydrobenzoic acid derivatives [32,47]. The stalk is composed of fibers such as cellulose, hemicellulose and lignin and rich in tannins, especially procyanidins, although they may also contain stilbenes, flavonols, hydroxycinnamic acids and flavano-3-ols [22,32]. Finally, grape seeds contain oils (mixture of saturated and unsaturated fatty acids), oligosaccharides, protoanthocyanidins and flavano-3-ols ((+)-catechin, (−)-epicatechin, procyanidin B1 and procyanidin B2) [22,32,47].

The phenolic content is dependent on several factors, namely the type of grape, the variety, the way they are processed, the maturation content and the environmental and agronomic conditions [32]. The bagasse from the red winemaking is a fermented pomace with a low sugar content and, in general, a low phenolic content. If it comes from white winemaking, the bagasse is not fermented and is potentially richer in sugars and phenolic compounds.

Bioactive compounds extracted from grape pomace have been shown to possess biological activities, such as anti-aging, the ability to protect cells from UV damage, and the promotion of hair growth [40,43,52,53,54,55], antioxidant [49,51,55,56], anti-inflammatory [49,51], antimicrobial [56,57], antiglycant [58], anticancer and cytotoxic activities [56], as well as having beneficial effects both at the hepatic [59] and cardiovascular level [50]. Therefore, extracts from grape pomace are attractive for applications in the pharmaceutical [49,50,51,58,59,60], food [61,62,63,64,65] and cosmetic [34,55,66,67,68] industries.

Several applications have already been developed for the recovery of waste and by-products from the winery industry (Figure 8), but, in general, they are not economically viable, as they require substantial amounts of resources and energy for their correct management. The most used disposal methods for grape pomace are landfilling, incineration and composting. In addition to these, a small percentage is used in distillation processes, to produce liquors, fertilizers or animal feed. Several industries also try to produce pasta and flour from this by-product. However, there are drawbacks to the current solutions since grape pomace contains anti-nutritive and antimicrobial compounds that can negatively affect plant cultures and the digestion process of animals [21,24,32,69]. One could think of incinerating or disposing these residues in the soil; however, the phenolic compounds give the bagasse an acidic pH that makes the soil more resistant to biological degradation. Furthermore, improper disposal leads to the contamination of groundwater and surface water [69]. Thus, there are still challenges, both economic and environmental, to achieve a circular economy within the wine industry [69].

In recent years, numerous alternatives of exploring grape pomace have been studied, all with the aim of extracting and valorizing the bioactive compounds present in this by-product, aiming at the maximum use of all raw materials and by-products and reducing the production of waste.

## 3. Extraction of Bioactive Compounds

Several techniques can be used for the extraction of bioactive compounds. The traditional techniques encompass methods such as maceration, percolation and Soxhlet. These techniques use harmful solvents, such as methanol, ethanol, propanol, acetone and ethyl acetate [70,71], are non-selective and normally use high temperatures with a high energetic cost [70]. In 2012, the green extraction of natural products was defined based on the definition of green chemistry as follows: “Green Extraction is based in the discovery and design of extraction processes which will reduce energy consumption, allows use of alternative and renewable natural products, and ensure a safe and high-quality extract/product” [72]. Green extraction techniques were developed throughout the years [71], such as enzyme-assisted extraction (EAE), fermentation-assisted extraction (FAE), mechanochemically assisted extraction (MCAE), extrusion, ultrasound-assisted extraction (UAE), microwave-assisted extraction (MAE), liquefied gas technology (LGE), pulsed electric field (PEF) technology, high-voltage electrical discharge (HVED), ohmic heating, infrared and ultraviolet light techniques, solar-based techniques, supercritical fluid extraction (SFE) and subcritical water extraction (SWE) [72]. These techniques emerged to replace conventional techniques and are known for the almost total recovery of bioactive compounds from plant matrices, with the use of a reduced volume of organic solvents, for the low energy consumption and for their short extraction time.

Among the various green extraction techniques recently developed, SWE can be considered one of the most promising ones. Therefore, in the following section, the main principles behind this technique, as well as its advantages and drawbacks, are discussed. In addition, the SWE of bioactive compounds from grape biomass is addressed as the focus of this review.

### 3.1. Subcritical Water Extraction

SWE is an extraction technique that consists of changing the polarity of water by adjusting the pressure and temperature to selectively extract compounds of different polarities from plant matrices [73,74]. The use of water as a solvent has limitations due to its high dielectric constant and consequent polarity at 25 °C, due to its ability to establish strong hydrogen bonds. As such, water is considered an inappropriate solvent for low-polarity compounds [73,75,76,77]. This is important, since in general, the extraction of nonpolar compounds requires the application of nonpolar solvents that are mostly toxic and prohibited in food and pharmaceutical products, for example [74]. However, this obstacle can be overcome by using subcritical water [74,76]. Water is in its subcritical state when it is maintained between 100 °C and 374 °C (its critical temperature) and between 1 MPa and 22.1 Mpa (its critical pressure). In these conditions, water is maintained in the liquid state [73,74,76,77,78,79,80,81,82,83,84].

When temperature and pressure are increased, significant changes occur in the physical and chemical properties of water, namely an increase in diffusivity and a constant decrease in its viscosity, surface tension and dielectric constant, with the latter resulting from the weakening of hydrogen bonds [78,79,80,81,84]. All these changes in the physical and chemical properties of water allow this type of extraction to be more efficient than commonly used techniques, due to the faster mass transfer, as well as the better impregnation of the extraction solvent in the plant matrix [84]. At 25 °C and atmospheric pressure, the dielectric constant of water is 80 [76]. If the temperature is increased to, for example, 250 °C and the pressure is increased to 25 bar, the dielectric constant drastically decreases to 25, being in the range of the dielectric constants of organic solvents, at room temperature (Figure 9). Thus, at these temperatures and pressures, water becomes capable of extracting compounds of medium to low polarity, similarly to organic solvents [73,75,76,77,80,81,82].

Figure 10 shows a schematic of a subcritical water extractor. The equipment consists of the reaction vessel, to which the solvent and the plant sample are added, which is placed inside a heating mantle and closed with the help of a clamp. The reaction vessel also contains cooling water inside, the stirrer rod and turbines to ensure the stirring of the mixture. The reaction controller allows parameters to be regulated, such as temperature, pressure and rotations. In addition, an inert gas, usually nitrogen, can also be used to maintain an inert atmosphere [79,80,85].

There are two types of equipment (Figure 11), static (discrete mode) and dynamic (continuous flow), which can be used individually or together. In the static extraction process, the raw material is moistened with water at room temperature, and only then is it subjected to high pressures and temperatures. The prolonged exposure time to elevated temperatures can cause degradation of the thermolabile compounds to be extracted, which is a disadvantage. The dynamic extraction process is a continuous process, which consists of introducing the raw material into the extraction vessel, which is continuously supplied with heated water, with the help of a pump. This mode allows the extraction efficiency to be increased as it increases the mass transfer and decreases the extraction time. However, the final extract may have to undergo a concentration step. Thus, an extraction in the static-dynamic mode can also be performed, through which more concentrated extracts are obtained with a shorter extraction time [85,86,87].

The subcritical extraction process occurs through diffusion and convection processes, resulting in the mass transfer from the vegetal matrix to the extraction solvent. The energy that comes from the high temperatures leads to the breakdown of adhesion interactions between the solute and the plant matrix and the cohesive solute–solute interactions. This occurs because there is a decrease in the activation energy necessary for the desorption process to occur. Simultaneously, high pressures more effectively force water to penetrate the plant matrix through the pores to what would occur under ambient pressure [76,80,84].

Temperature, pressure, particle size, solvent flow rate and the addition of co-solvents are some examples of factors that affect the efficiency of subcritical water extraction [76,80,88]. Among these, temperature is the most crucial factor in a subcritical water extraction process given its enormous influence on both the efficiency and the selectivity of the extraction itself, since it directly modifies the solvation capacity of water [76,78,81,84,85,89]. The water molecule becomes less polar with the breaking of hydrogen bonds caused by this increase in temperature. This can be an advantage as far as it solves the dielectric constant problem, decreasing its value and making water a solvent capable of solubilizing compounds with moderate or low polarity, or even non-polar compounds [78,80,85]. In addition, the surface tension and viscosity of water also decrease, contrary to diffusivity. This promotes the penetration of the solvent into the matrix, increasing the mass transfer from the plant to the subcritical water and consequently increasing the efficiency and speed of extraction compared with other extraction techniques [76]. Finally, high temperatures lead to the breaking of van der Waals bonds, hydrogen bonds and dipoles, which decreases the activation energy required for the desorption process [76]. However, an increase in temperature is not always advantageous. This is a factor that can also cause the thermal degradation of compounds that are thermolabile and promote oxidation and hydrolysis reactions that will alter the target compounds [74]. Still, hydrolysis reactions can also facilitate the breakdown of cell walls, facilitating the release of bioactive compounds present inside the plant cell [74]. Considering the previous information, it can be concluded that the optimization of the extraction temperature is extremely important for the success of the subcritical water extraction implementation. For example, Yang et al. [90] used temperatures between 200 and 250 °C and verified that they could not extract terpene from basil and oregano samples, causing its degradation. However, after increasing the temperature above 300 °C, they were able to increase the extraction efficiency due to the decrease in the dielectric constant of water and the consequent increase in the solubility of non-polar compounds [90].

An adjustment in pressure also changes the physical state of the water. In the subcritical process, in general, the pressure is maintained in the range of 10–80 bar to ensure that water remains in the liquid state even at high extraction temperatures. However, changes in pressure do not significantly affect the extraction process if the water remains in the liquid state [76,80,81,89]. High pressures allow the penetration of the solvent through the pores of the matrix, which would not be possible under ambient pressure [76,81,85].

The size of the particles is another parameter that must be considered during extraction, since, in general, smaller particles allow better extraction yields with smaller extraction times to be obtained, as there is an increase in the surface area of contact between the solute and the extraction solvent when compared with the use of bigger particles [76,80].

In the case of extractions in dynamic equipment, an increase in the amount of solvent supplied to the extraction system has also been shown to increase the efficiency of the reaction and decrease the residence time. However, if the solvent flow rate is excessive, the compounds may be diluted and require an additional concentration step, which is undesirable [80]. The addition of co-solvents that increase the solubility of low-polarity substances also affects the physical and chemical properties of water, such as the dielectric constant, surface tension and diffusivity. Consequently, the addition of these new substances can affect the efficiency of the reaction, so their nature, interactions and amount must be better understood [76].

The advantages of this extraction method are its simplicity and the fact that it only uses water as a solvent, which makes the extraction technique green and lowers the associated cost. Polar, moderately polar and non-polar compounds can be separately extracted using this technique just by varying pressures and temperatures, which is a great advantage, for example, compared to supercritical extraction. In addition, the technique has a short extraction time and high efficiency and enables a continuous process [80]. The high efficiency is due to the use of higher temperatures than those reported for most techniques, and to the hydrolysis reactions that break the cell walls and facilitate the release of compounds [74]. However, this technique also has disadvantages, namely the difficulty in separating the bioactive compounds from the extracts, the thermal degradation and oxidative damage that can occur when using high temperatures, and the difficulty in cleaning the equipment [80,91]. In addition to these disadvantages, undesired products may be created due to damage in the plant material, allowing the release of bioactive compounds that were inside them into the water. These compounds can interact with each other, forming new compounds that logically have new properties. For example, samples containing substantial amounts of sugars at elevated temperatures can generate Maillard reactions and caramelization with the formation of new compounds [80,91].

### 3.2. Subcritical Water Extraction of Grape Biomass

The subcritical extraction technique was considered efficient to extract bioactive compounds [76], namely phenolic compounds [92,93,94,95,96,97,98], terpenes [93,99,100,101,102], flavonoids [103,104], anthocyanins [105], polysaccharides [96,97,106,107,108], proteins [109,110], biopolymers [111], tannins [97], fibers [98] and amino acids [110]. Recently, extraction methods have emerged using deep eutectic solvents (DESs) [77,112,113,114] as extraction co-solvents. In the last two decades, several studies have arose that use the subcritical water extraction method to extract bioactive compounds from by-products from the wine industry.

In 2006, Garcia-Marino et al. [115] used grape seeds from Tempranillo grape pomace (*Vitis vinifera* L.), previously frozen at −35 °C. The samples were lyophilized and crushed in powder form, in which individual and sequential extractions were performed. Extraction temperatures ranged from 50 °C to 150 °C, maintaining a pressure of 103,421 bar and a constant nitrogen purge. For comparison, another extraction was carried out (mixture methanol/water 75/25% *v*/*v*, homogenization, ultrasonic bath, 15 min). The most efficient extraction was the sequential one, lasting 90 min at temperatures of 50 °C, 100 °C and 150 °C, with a total polyphenol content (TPC) extracted of 582.5 mg/100 g of dry weight (dw) compared to 292.7 mg/100 g dw through the methanol extraction process. The single extraction at 150 °C was also more efficient than the conventional method, with a TPC of 380.6 mg/100 g dw. The subcritical water extraction method allowed the extraction of several catechins and proanthocyanidins, as well as gallic acid, in a total of 25 compounds identified using HPLC-DAD-MS (high-performance liquid chromatography coupled to a diode array detector and mass spectrometer). The authors concluded that the sequential extraction technique would be the most adequate to obtain larger amounts of phenolic compounds compared to the conventional method, with subcritical water being considered by the authors as a good solvent for the extraction of flavanols.

Three years later, Monrad et al. [116] used seeds and skins of Sunbelt red grapes (*V. labrusca *L.) that were frozen at −20 °C. The sample was lyophilized, ground to a fine powder and frozen at −70 °C. The sample and the hydroethanolic solvent were placed in a reaction vessel with cellulose paper inside. The extraction conditions used were a pressure of 6.8 MPa and 90 s of nitrogen purge. Various percentages (*v*/*v*) of ethanol in water (10%, 30%, 50% and 70%) and six different temperatures (40 °C, 60 °C, 80 °C, 100 °C, 120 °C and 140 °C) were tested. For comparison, a conventional extraction was performed with a mixture of methanol/water/formic acid (60/37/3, *v*/*v*/*v*). Fourteen anthocyanins were identified using HPLC-ESI-MS (high-performance liquid chromatography-electrospray ionization tandem mass spectrometer) and extracted by most of the solvents used. The authors found that 70% and 50% ethanol (*v*/*v*) extracted more anthocyanins, with a total average of 463 and 455 mg/100 g dw, respectively. Regarding temperatures, more anthocyanins were extracted at 100 °C (450 mg/100 g dw), 80 °C (436 mg/100 g dw) and 120 °C (411 mg/100 g dw). Among all the conditions, 100 °C and 50% ethanol allowed the extraction of the highest amount of total anthocyanins (497.43 ± 13.54 mg/100 g dw), a value higher than that obtained using the conventional method (442.88 ± 15.29 mg/100 g dw). The same authors [117] tested the same conditions, but adding sand as a dispersing agent. Several percentages (*v*/*v*) of ethanol in water (0%, 10%, 30%, 50%, 70% and 90%) and six different temperatures (40 °C, 60 °C, 80 °C, 100 °C, 120 °C and 140 °C) were tested. For comparison, a conventional extraction was performed with a mixture of acetone/water/acetic acid (70/29.5/0.5, *v*/*v*/*v*). Eight procyanidins were identified using HPLC-ESI-MS. In this case, a temperature of 120 °C and a percentage of 70% ethanol allowed a more efficient extraction to be obtained, reporting a total value of procyanidins of 5712 ± 217 mg/100 g dw compared to 4955 ± 275 mg/100 g dw obtained through conventional extraction.

Two years later, Monrad et al. [118] used white grape pomace from the Zinfandel variety (*Vitis vinifera* L.) with 58% moisture. The bagasse was frozen at −20 °C and a part underwent a drying process at 40 °C until reaching 4% humidity. The dried sample was ground for 30 s and frozen again at −20 °C. The wet bagasse was crushed immediately before extraction for the same amount of time. A semi-continuous system, with a nitrogen purge and a pressure of less than 4.1 bar, was used. The sample mass used varied between 5 and 25 g, and the extraction time and temperatures were 28 min, 17 min and 11 min and 60 °C, 100 °C and 140 °C, respectively. For comparison, a conventional extraction was performed using a mixture of methanol/formic acid/water (60/37/3, % *v*/*v*/*v*) for anthocyanin extraction and a mixture of acetone/water/acetic acid (70/29.5/0.5, % *v*/*v*/*v*) for procyanidin extraction. The samples were homogenized with the solvents for 30 s at room temperature. Among the tested temperatures, 140 °C allowed, in general, a higher extraction yield. Regarding the dried samples, the conditions that allowed a larger extraction yield of anthocyanidins (83.6 mg/100 g dw) and procyanidins (2372.2 mg/100 g dw) were 5 g of sample, a flow rate of 5 mL/min, 28 min and 140 °C, and 5 g of sample, a flow rate of 5 mL/min, 28 min and 60 °C, respectively. From the non-dried samples, a larger amount of anthocyanidins (119.5 mg/100 g of dw) and procyanidins (2607.0 mg/100 g dw) was extracted using the following conditions: 5 g of sample, a flow rate 15 mL/min, 11 min and 140 °C, and 25 g of sample, 5 mL/min, 28 min and 140 °C for anthocyanidins and procyanidins, respectively. Traditional methods extracted close amounts of procyanidins (2464.0 mg/100 g dw), but slightly higher amounts of anthocyanidins (135.6 mg/100 g dw), for the wet bagasse. Eight procyanidins and fourteen anthocyanins were identified using HPLC-ESI-MS and HPLC-MS, respectively.

Still in 2012, Aliakbarian et al. [82] used Croatina grape pomace, which was dried at 40 °C for 24 h. Afterward, the sample was ground for 20 s and frozen at −20 °C. The extraction was carried out with water either in static mode for 30 min, at a predetermined temperature, constant flow rate and closed pressure valve, or in continuous mode for 100 min, with a flow rate set between 1 and 2 mL/min and opening and closing the pressure valve. To compare, a conventional extraction was performed with pure ethanol or ultrapure water and placed on a magnetic stirrer for 19 h at 25 °C. The highest TPC (32.49 ± 2.63 mg GAE/g dw, GAE—gallic acid equivalents) and flavonoids (15.28 mg ± 1.02 CE/g dw, CE—catechin equivalents) were obtained at 140 °C with pressures of 8 MPa and 11.5 MPa, respectively. The largest anti-radical power (13.85 ± 1.26 μg DPPH/μL extract) determined using the DPPH method (2,2-diphenyl-picrylhydrazyl radical) was obtained at 15 MPa and at 140 °C. The TPC and total flavonoid content (TFC) obtained were higher than those obtained using the conventional method, either using water as a solvent (1.72 ± 0.05 mg GAE/g dw and 1.25 ± 0.05 mg CE/g dw) or using ethanol (7.87 ± 0.48 mg GAE/g dw and 14.49 ± 2.17 mg CE/g dw). However, the same is not true for the determination of the anti-radical power in which the conventional extraction allowed a superior antioxidant capacity to be obtained: 4.19 ± 0.50 and 22.57 ± 1.87 μg DPPH/μL extract for water and ethanol, respectively.

In 2014, Rajha et al. [119] used grape pomace from the Cabernet Sauvignon variety. A part of the sample underwent a drying process at 45 °C and the rest was analyzed while wet and subsequently ground. The extraction process took 15 min and was carried out at six different temperatures (40 °C, 60 °C, 80 °C, 100 °C, 120 °C and 140 °C), for three different ethanol/water ratios (30%, 50% and 70%) at a pressure of 100 bar, undergoing a nitrogen purge for 120 s. The highest levels of phenolic compounds were obtained at a temperature of 140 °C with a ratio of 70% ethanol/water, with the wet bagasse (16.2 g GAE/100 g dw) having a higher yield than the dry bagasse (7.28 g GAE/100 g dw).

In 2015, Duba et al. [120] used grape pomace from the Pinot Nero variety, from which they separated the seeds from the grape skins. The mixture was frozen at −20 °C and subjected to a drying process at 55 °C for 48 h. Seeds and hulls were separated using sieves, cleaned and stored in the dark at room temperature. Following that, both samples were crushed immediately before the extraction process. In addition to these steps, the seeds underwent an additional step of degreasing with supercritical carbon dioxide (50 MPa, 50 °C, 8 g/min). The sample was added to the reaction vessel, and for 15 min, nitrogen was directly placed in the vessel and inside the vessel for an additional 5 min. The extraction was performed for 2 h in static mode, with extracts collected every 20 min. Regarding the peels, the conditions that allowed greater TPC values were 120 °C, 2 mL/min, 10 MPa and 120 min, and the content in the peels was lower than that of the seeds (77 ± 3 and 124 ± 1 mg GAE/g, respectively).

In 2017, Tian et al. [94] used outdoor dried grape seeds, which were pulverized and sieved. The powder was stored in a desiccator at room temperature. In a serial extraction process, the sample and ultrapure water were added to the reactor, followed by a nitrogen purge. For comparative purposes, three other types of extractions were performed: extraction through reflux (mixture ethanol/water 75/25% *v*/*v*, water bath at 70 °C for 6 h), ultrasound (mixture ethanol/water 40/60% *v*/*v*, 12 min at 50 KHz) and microwaves (mixture ethanol/water 60/40% *v*/*v*, 15 min at 500 W at 50 °C). SWE (150 °C, 25 min, 1 MPa) allowed the extraction of a higher amount of resveratrol (6.90 ± 0.03 μg/g material) compared to the extraction through reflux (4.16 ± 0.2 μg/g material), ultrasound (3.42 ± 0.26 μg/g material) or microwaves (4.66 ± 0.25 μg/g material).

In the same year, Pedras et al. [121] used white grape marc, frozen with liquid nitrogen and lyophilized for 3 days. The bagasse was left at room temperature, ground until obtaining a powder and frozen at −18 °C. At a pressure of 100 bar, three different temperatures (170 °C, 190 °C and 210 °C) and flow rates (5–10 mL/min) were tested. Four extracts were collected at different ranges of temperature, T < 50 °C, T = 50–130 °C, T = 130–210 °C and T = 210 °C, after which they were stored at 4 °C and lyophilized. For comparison purposes, a conventional extraction was performed (mixture ethanol/water (25/75% *v*/*v* at 50 °C)) for 18 h at 150 rpm. A temperature of 210 °C and a flow rate of 10 mL/min allowed the extraction of a larger amount of phenolic compounds (TPC = 113.4 ± 5.1 mg/g extract) compared with the value obtained using the conventional extraction (47.3 ± 1.4 mg/g extract) for an extraction of 180 min.

In 2018, Moreira et al. [92] used firewood from pruning vines from Tinta Roriz (TR) and Touriga Nacional (TN) varieties (*Vitis vinifera* L.) that were dried at 50 °C for 24 h, crushed (<1 mm) and stored at room temperature. For comparison purposes, the same authors tested a microwave-assisted extraction (MAE) (mixture of ethanol/water 60/30% *v*/*v* for 20 min at 100 °C) and a conventional extraction (mixture of ethanol/water 50/50% *v*/*v* for 2 h at 55 °C in a water bath with stirring). SWE was carried out through a nitrogen purge at a pressure of 40 bar and at 150 °C for 40 min under stirring at a frequency of 3 Hz. The TPC for the TN variety for SWE was 1502 mg/100 g dw, higher than the value obtained through conventional extraction (360 mg/100 g dw) and MAE (1228 mg/100 g dw). In the case of the TR variety, the SWE (1142 mg/100 g dw) was less efficient than the MAE (1421 mg/100 g dw), but more efficient than the conventional extraction (501 mg/100 g dw).

In 2019, Kashtiban et al. [122] used grape skins of the Siah-Sardasht variety that were frozen at −20 °C, to which an ultrasonic pretreatment was performed at a frequency of 21 KHz, at 400 W and with a titanium probe (14 mm) for 30 min at 50 °C. The extracts obtained were transferred and used for SWE, which was performed using ultrapure water, with the passage of nitrogen. Two different pressures (20 bar and 40 bar), two different temperatures (100 °C and 150 °C) and two different times (15 min and 30 min) were tested. For comparison, a conventional extraction was performed using grape skin and methanol, at 200 rpm and at room temperature for 20 min. The extraction conditions that allowed for greater efficiency were 40 bar, 150 °C and 30 min for TPC (1956.52 ± 8.54 mg/mL) and TFC (155.4 ± 4.22 mg/L), and for the antioxidant activity determined using the DPPH method (93.95 ± 1.97 M). These values were all higher than those obtained using the conventional extraction method: TPC (1112.71 ± 0.45 mg/mL), TFC (96.67 ± 1.33 mg/L) and DPPH (74.55 ± 0.67 M).

In 2020, Yammine et al. [123] used grape pomace from four varieties (Chardonnay, Cabernet Franc, Merlot and Dunkelfelder). The brands were pressed at 2105 Pa and the white variety (Chardonnay) was frozen at −20 °C, while the three red varieties were pre-treated with 50 mg of SO_2_/kg of marc before freezing. Grape seeds and skins were separated through sieving. Each extraction was performed using a ratio of 5/1 (liquid/solid) with deionized water. A pressure of 2105 Pa was chosen, and a flow rate of 6 mL/min was used with three different temperatures (100 °C, 150 °C and 200 °C). To compare, a conventional extraction was performed (a mixture of ethanol/water 50/50% *v*/*v* in a liquid/solid ratio of 5/1 at 160 rpm and 420 min). The temperature of 200 °C allowed a greater extraction in almost all varieties as well as compared with the remaining SWE temperatures and with conventional extraction. From the different varieties analyzed, Dunkelfelder was the richest in terms of bioactive compounds. For this variety, the values obtained using SWE were always higher than those obtained using the conventional method, with the following results obtained using SWE for total proanthocyanidins (72.52 ± 2.43 mg tannins/g dw), TPC (94.78 ± 0.49 mg GAE/100 g dw) and TFC (198.86 mg/100 g dw).

In the same year, Dorosh et al. [124] used firewood from pruning vines of the TR and TN varieties (*Vitis vinifera* L.), which they had previously tested [92]. SWE was used at two different temperatures (125 °C and 250 °C), with pure water for 50 min at 250 rpm. To evaluate the extraction efficiency, TPC, TFC and antioxidant activity determined using the DPPH and FRAP methods (Ferric reducing antioxidant power) were assessed and it was verified that the best results were obtained at 250 °C. For TN, the following values were obtained: 165 ± 8 mg GAE/g dw, 46 ± 3 mg EE/g dw (EE—Epicatechin equivalents), 202 ± 22 mg TE/g dw and 186 ± 21 mg AAE/g dw. In the case of the TR variety, the reported levels were higher than those of the TN variety: 181 ± 12 mg GAE/g dw, 51 ± 6 mg EE/g dw, 203 ± 22 mg TE/g dw and 202 ± 14 mg AAE/g dw.

Also in 2020, a study using DESs was reported by Loarce et al. [113] using grape pomace (*V. vinifera* L. *cv.*) from the Tempranillo variety. The sample was lyophilized at 80 °C under a vacuum of 1.1 × 10^−2^ mbar, ground and then frozen at −20 °C. For the SWE, 2 g of lyophilized bagasse and 1 g of diatomaceous earth (dispersing agent) at 10.34 MPa and a nitrogen purge for 80 s were used, with different temperatures ranging from 40 °C to 120 °C. Eight DESs were used (choline chloride/oxalic acid (1:1); choline chloride/lactic acid (1:2); choline chloride/fructose/water (2:1:1); choline chloride/ethylene glycol (1:2); choline chloride/1,2-propanediol (1:2); choline chloride/urea (1:2); citric acid/maltose/water (4:1:5); and citric acid/fructose/water (1:1:2)) in four DES/water ratios (10, 20, 30 and 40, % *v*/*v*). After the SWE, water was added to the obtained extracts, and they were subsequently frozen at −18 °C. The authors concluded that the 30% DES choline chloride/urea (1:2) at 100 °C allowed for a better extraction efficiency. Under these conditions, 208.97 ± 12.17 mg/g of catechins, 15.99 ± 2.81 mg/g of tannins, 0.14 ± 0.02 mg/g of hydroxycinnamic acids and 0.13 ± 0.03 mg/g of flavonols were extracted. These authors also determined the antioxidant activity using ABTS (2,2′-azino-bis-(3-ethylbenzothiazoline-6-sulfonic acid, 0.74 ± 0.11 mM Trolox/g) and DPPH (0.50 ± 0.03 mM Trolox/g) methods. The obtained data were compared with those obtained using only water in the SWE, and it was possible to verify an increase in the extraction efficiency through the addition of DES. In the SWE with water alone, 16.80 ± 5.91 mg/g of catechins, 1.52 ± 0.23 mg/g of tannins, 0.07 ± 0.01 mg/g of hydroxycinnamic acids and 0.02 ± 0.00 mg/g of flavonols were extracted. In the case of antioxidant activity, values of 0.17 ± 0.00 mM/g and 0.11 ± 0.01 mM/g were obtained for the ABTS and DPPH methods, respectively.

Finally, in 2022, Barriga-Sánchez et al. [125] used grape pomace from the Quebranta variety (*Vitis vinifera*). The bagasse was dried at 25 °C for 36 h until reaching 13% moisture. The seeds were separated, dried at 40 °C for 6 h until 7% moisture content, crushed and sieved. Both samples were stored in vacuum bags and protected from light at 5 °C. Some seeds were pre-treated with supercritical carbon dioxide (degreasing). The SWE was carried out at 120 °C and 100 bar, using deionized water as the solvent and five layers of glass beads (5 mm for a total of 700 g). Each layer of glass beads was interspersed with one layer of seeds with pre-treatment, and the last layer was just seeds without pre-treatment. Water was added to the reactor with a pump for 40 min at 15 mL/min. These conditions were maintained for 3 h (static mode). For comparison, three maceration processes were carried out (mixture ethanol/water 70/30% *v*/*v*, methanol and mixture acetone/water 50/50% *v*/*v*). In terms of TPC, SWE (167.56 ± 10.40 mg GAE/g dw) was more efficient than maceration with 70% ethanol (27.89 ± 2.24 mg GAE/g dw), methanol (32.40 ± 2.15 mg GAE/g dw) and 50% acetone (40.43 ± 3.91 mg GAE/g dw). In terms of antioxidant activity, determined using the DPPH and FRAP methods, SWE was also more efficient (1479.90 ± 12.86 and 845.13 ± 95.32 μmol TE/g dw) than maceration with 70% ethanol (179.59 ± 46.36 and 200.39 ± 19.86 μmol TE/g dw), methanol (174.74 ± 26.18 and 253.19 ± 10.89 μmol TE/g dw) and 50% acetone (409.82 ± 81.30 and 347.07 ± 36.55 μmol TE/g dw).

## 4. Final Remarks and Future Perspectives

SWE can be considered a green extraction technique following some of the principles of green chemistry and the principles of green extraction for natural products since it uses a safe solvent (water) and the extraction process can be considered safe, robust and controlled, performed in a single unit operation. The term green chemistry (GC) was first defined by Paul Anastas in 1991 as the utilization of a set of principles that reduces or eliminates the use or generation of hazardous substances in the design, manufacture and application of chemical products, and it comprises 12 principles defined in 1988 by Paul Anastas and John Warner [126]. Within these 12 principles are specified ways to decrease chemical production in the environment. GC in many cases involves the re-design of conventional routes to find optimal conditions, maximizing performance and profit [127]. The concept of green extraction of natural products arouses the need to demonstrate how to perform the extraction of natural products on a laboratory scale and industrial scale in order to have an optimal consumption of raw materials, solvents and energy [72].

SWE has been shown to be a suitable alternative for the extraction of bioactive compounds from food waste with several advantages: it is simple, can extract from polar to non-polar components, has a short extraction time, has a high efficiency and enables a continuous process. However, reported research from SWE still presents some drawbacks, such as most of the studies being performed at the laboratory scale. Future steps from SWE should include the scaling up to the industrial level and the design of industrial equipment. Indeed, some results at the pilot scale have demonstrated the potential development of large-scale SWE processes, bearing in mind the principles of GC [128]. Further, the sustainable scaling up of the SWE process would significantly contribute to the understanding, advancement and future applications of natural extracts obtained with SWE in dealing with health problems, as the subcritical water extracts can be directly used in pharmacological and toxicological tests.

Despite the SWE laboratory-scale process being widely explored, there is still room for improvement and the addition of a co-solvent can improve the extraction, increasing the solubility of low-polarity compounds and affecting water properties such as the dielectric constant, surface tension and diffusivity. Neoteric solvents are good candidates to be used as co-solvents of SWE. Neoteric solvents have several interesting properties: they have a negligible vapor pressure, they are chemically and physically stable, and their polarity, hydrophilicity and solvent miscibility can be tunned depending on the intended application [129,130]. Neoteric solvents can be grouped into supercritical fluids, fluorinate solvents, ionic liquids, switched solvents, thermomorphic solvents, solvents derived from biomass, liquid polymers and DESs [130,131,132,133,134,135,136].

DESs have already been used in extraction techniques, such as UAE and MAE [71]. It has been proven that the extraction is improved with the use of DESs as co-solvents but that the extraction capabilities are highly dependent on the physicochemical composition of the DESs and the composition of the target compounds [71]. Nevertheless, this class of solvents seems highly efficient for bioactive compound extraction. As referred to in the former section, DESs were used only in one work as co-solvents in SWE for grape biomass extraction [113]. The authors compared the SWE performed with different percentages of DESs/water and only with water, and it was possible to verify an increase in the extraction efficiency through the addition of DESs in all the parameters studied [113]. There is still a great amount of work to be developed to explore the capability of DESs as co-solvents in SWE; however, the obtained results are promising and represent a step toward the circular economy.

There has been continuous research in GC with emphasis on new metrics, such as the atom economy (AE), process mass intensity (PMI) and new business models, for instance, the implementation of a life cycle assessment (LCA), that can be used for GC implementation [137]. It is not the objective of this review to give a comprehensive analysis of GC metrics; extensive reviews on this subject can be consulted for further information [127,138,139,140]. However, this is another area that should be explored for SWE to be able to gain a deeper understanding of the process as a sustainable way of extraction.

## Figures and Tables

**Figure 1 molecules-28-07538-f001:**
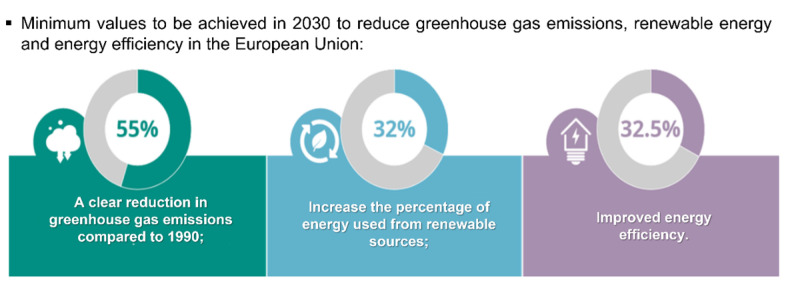
Targets for reducing European environmental problems. Adapted from [5].

**Figure 2 molecules-28-07538-f002:**
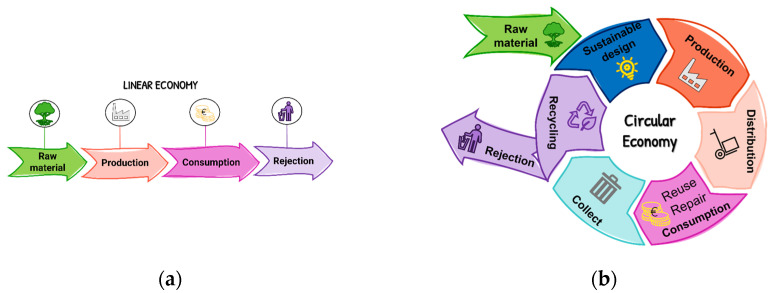
Representations of a linear (**a**) and circular (**b**) economy.

**Figure 3 molecules-28-07538-f003:**
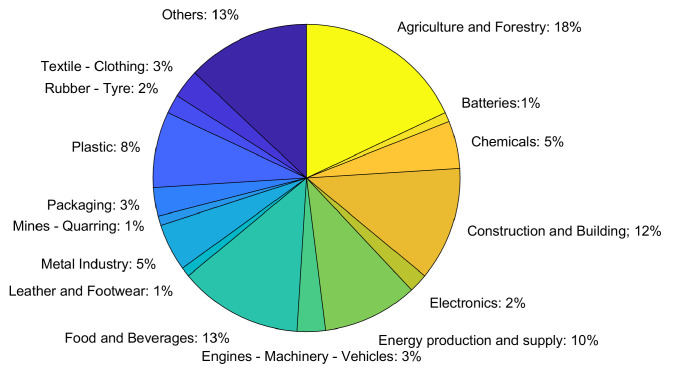
Representation of LIFE circular economy projects. Adapted from [7].

**Figure 4 molecules-28-07538-f004:**
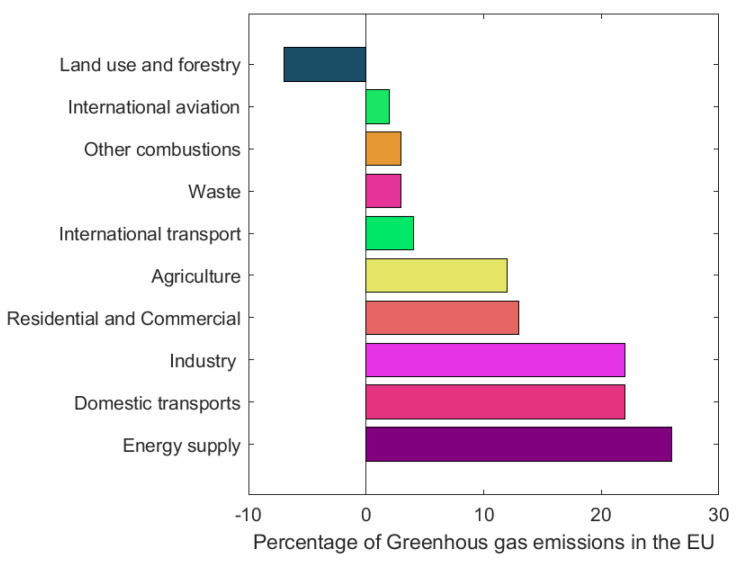
Sources of greenhouse gas emissions in the European Union in 2021. Adapted from [10].

**Figure 5 molecules-28-07538-f005:**
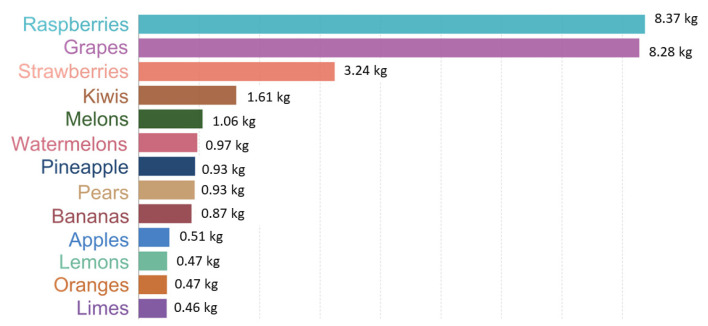
Greenhouse gas emissions per kilogram of fruit. Emissions are measured in carbon dioxide equivalents. Adapted from [9].

**Figure 6 molecules-28-07538-f006:**
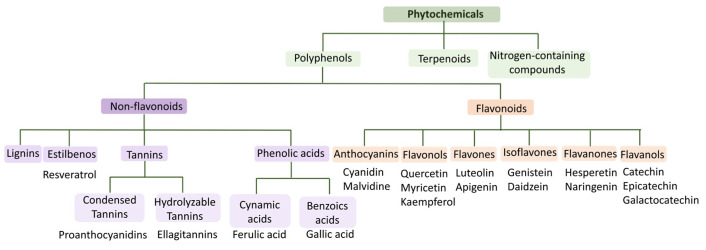
Phytochemical classification scheme and some examples. Adapted from [45,46].

**Figure 7 molecules-28-07538-f007:**
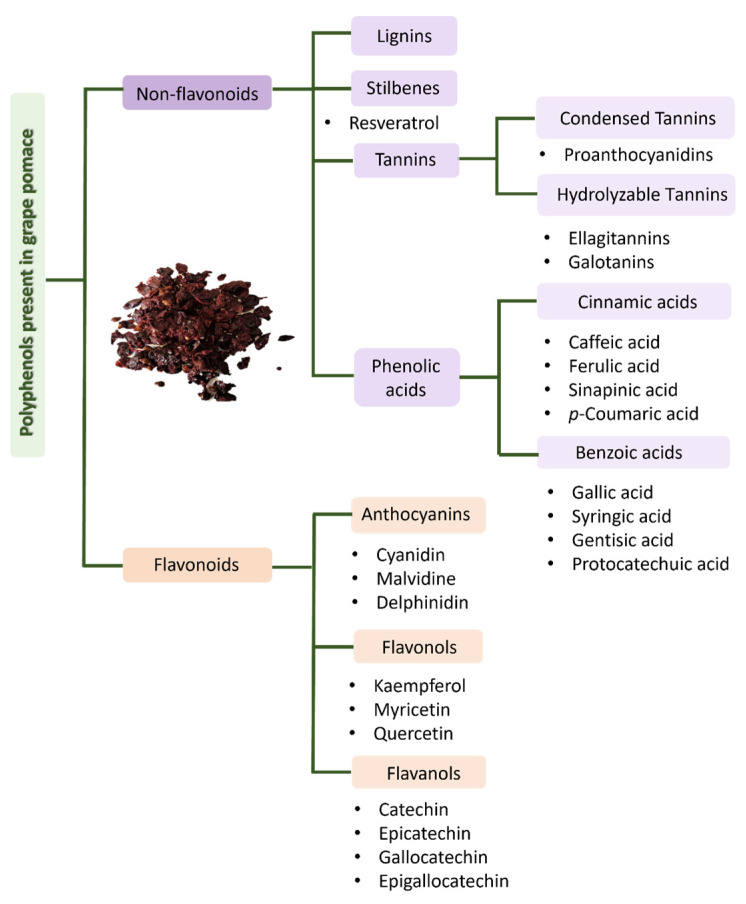
Major polyphenols from grape pomace.

**Figure 8 molecules-28-07538-f008:**
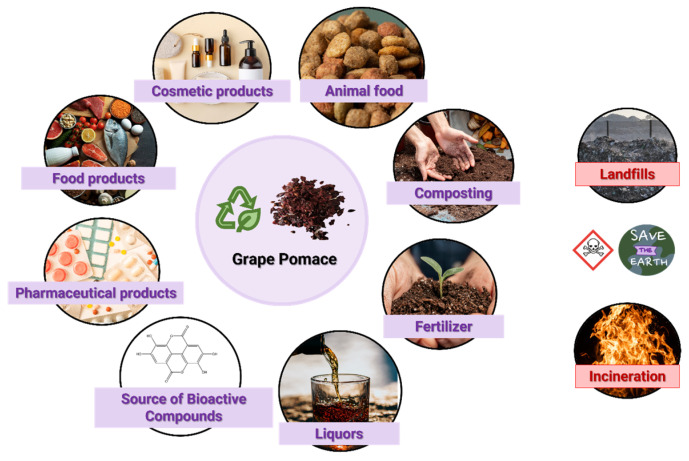
Different applications of grape pomace.

**Figure 9 molecules-28-07538-f009:**
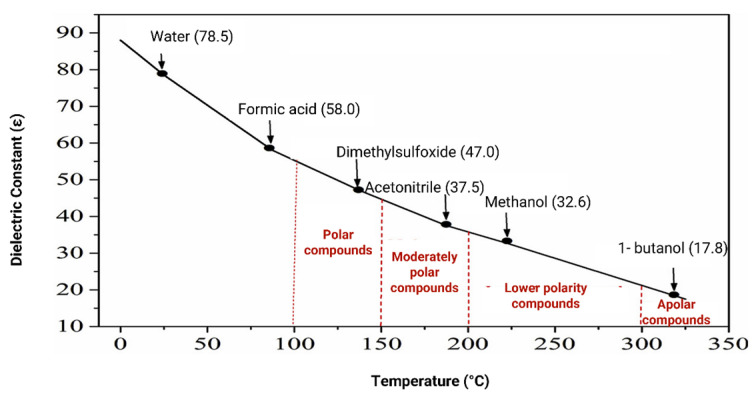
Variation in the dielectric constant of water with temperature at constant pressure (20 Mpa). Representation of the dielectric constant of solvents at 25 °C and 0.1 Mpa. With permission from Elsevier [80].

**Figure 10 molecules-28-07538-f010:**
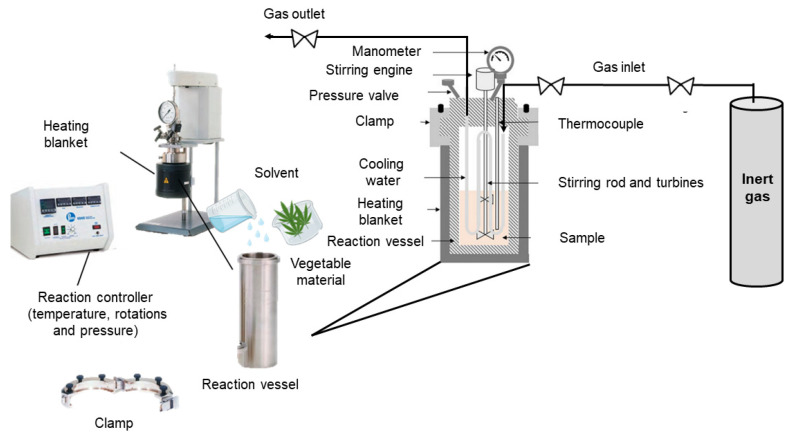
Scheme of a subcritical extractor. Adapted from [86].

**Figure 11 molecules-28-07538-f011:**
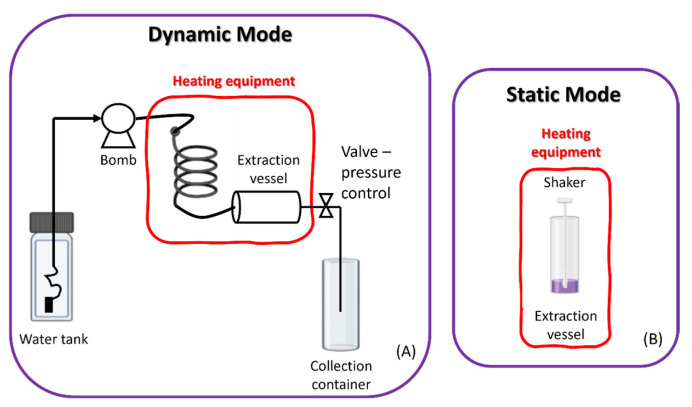
Subcritical equipment in dynamic (**A**) and static (**B**) mode.

## Data Availability

Data sharing not applicable.

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
