# Peer review of "Subcritical Water Extraction to Valorize Grape Biomass—A Step Closer to Circular Economy"

_molecules, 2023, doi:10.3390/molecules28227538_

Round 1

Reviewer 1 Report

Comments and Suggestions for Authors

In this work the authors present an overview of the extraction process by SWE (Subcritical Water Extraction) of grape biomass in a perspective of circular economy by valorization of the bioactive compounds extracted. The manuscript is well presented and well written but I believe that the bibliographical section needs greater integration precisely because it is a review.

Several works are reported in the literature in which methods of extracting nutraceuticals from biomass are reported but the authors have not reported (Benincasa, C.; Santoro, I.; Nardi, M.; Cassano, A.; Sindona, G. Eco-friendly extraction and characterisation of nutraceuticals from olive leaves. Molecules 2019, 24, 3481; Current Medicinal ChemistryVolume 19, Issue 23, Pages 4006 - 4013) . I believe that the manuscript can be accepted after minor revision.The authors can reported different methods of water extraction of nutaraceuticals.

Author Response

In this work the authors present an overview of the extraction process by SWE (Subcritical Water Extraction) of grape biomass in a perspective of circular economy by valorization of the bioactive compounds extracted. The manuscript is well presented and well written but I believe that the bibliographical section needs greater integration precisely because it is a review.

Several works are reported in the literature in which methods of extracting nutraceuticals from biomass are reported but the authors have not reported (Benincasa, C.; Santoro, I.; Nardi, M.; Cassano, A.; Sindona, G. Eco-friendly extraction and characterisation of nutraceuticals from olive leaves. Molecules 2019, 24, 3481; Current Medicinal ChemistryVolume 19, Issue 23, Pages 4006 - 4013) . I believe that the manuscript can be accepted after minor revision.The authors can reported different methods of water extraction of nutaraceuticals.

Reply: We thank the reviewer for the comments. We focus our review on the valorization of grape biomass in particularly with the use of subcritical water extraction. However, we thank the suggestions and the paper “Eco-friendly extraction and characterisation of nutraceuticals from olive leaves” was added to the introduction section.

Reviewer 2 Report

Comments and Suggestions for Authors

The authors presented a review on subcritical water extraction to valorize grape biomass, taking into consideration the aspect of sustainable development together with a circular economy approach.

The first paragraph introduces the reader to the reasons that lead to a necessary radical change in the entire world towards a circular economy approach and greater eco-sustainability of processes. Subsequently, data about production and exploitation of grape biomass together with its main bioactive compounds were shown to highlight the interesting characteristics of this biomass and the importance of recovering such compounds from it.

After a brief presentation of non-conventional extraction techniques, subcritical water extraction (SWE) was described, moving ultimately to the grape biomass SWE, which was presented in a comprehensive and careful manner.

 However, I do have some minor concerns that the authors should consider to improve the manuscript before publication. I would recommend a careful revision and editing of the text of the Abstract and Introduction.

Below are some suggested revisions:

Line 24: “has” instead of “as” (?)

Line 26: “can be” instead of “is”

Line 28-29: it may be important to cite also the “Green extraction” principles (see comments for Paragraph 3.1)

Line 36: remove “population” (repetition)

Line 40: “translated” instead of “translates”

Line 47: insert a fullstop or a semicolon after [2,5].

Line 50-52: 55% is related to the reduction of greenhouse gas emissions

Line 66: “demanded to explore new …..”

Line 69-71: “a circular economy would encourage” for example….(to remove repetition of “allow”)

Line 72-75: can you explain better this part?

Line 88: perhaps “important” instead of “importantly” (?)

Line 137-139: it may be important to cite the first 5/10 importing and exporting countries in the world.

Line 146: remove “and” after “thus,”.

Paragraph 2.1: lines 187-192 and lines 213-219, these two parts are connected, therefore it would be better to put them together and make a synthesis of them.

Line 210: remove this part “however,….used.”

Paragraph 3.1: in this part (line 249, line 256), as when the Green Chemistry was cited in the review, there is a lack of citations regarding the Green extraction concept and principles. Therefore, I would suggest including some references about this very important topic, which has paved the way for all non-conventional extraction techniques. I include some examples below:

-Farid Chemat, Maryline Abert Vian, and Giancarlo Cravotto. Green Extraction of Natural Products: Concept and Principles Int J Mol Sci. 2012; 13(7): 8615–8627.

-Green Extraction of Natural Products: Theory and Practice, First Edition. Edited by Farid Chemat and Jochen Strube. 2015 Wiley-VCH Verlag GmbH & Co. Chapter 1. Green Extraction: From Concepts to Research, Education, and Economical Opportunities, Farid Chemat, Natacha Rombaut, Anne-Sylvie Fabiano-Tixier, Jean T.Pierson, and Antoine Bily

-A review of sustainable and intensified techniques for extraction of food and natural products. Farid Chemat, Maryline Abert Vian, Anne-Sylvie Fabiano-Tixier, Marinela Nutrizio, Anet Režek Jambrak, Paulo E. S. Munekata, Jose M. Lorenzo, Francisco J. Barba, Arianna Binello and Giancarlo Cravotto Green Chem., 2020,22, 2325-2353

Line 385: remove repetition “production..products..”

Line 538: introduce a spacing between 3 and Hz

Line 562: mL instead of Ml

Paragraph 4:

Lines 632-644 Please, introduce some comments about Green extraction principles too.

Lines 642-647: this part is more suitable for the introduction, not for the final remarks

lines 662-664: this phrase is very similar to the one found in the ref. 131.

The authors could better underline what the future prospects of SWE are, and what are the shortcomings in understanding this extraction technique and applying it in a sustainable process.

Comments on the Quality of English Language

I would suggest a moderate editing of English language, especially for the Abstract and Introduction paragraphs.

Author Response

The authors presented a review on subcritical water extraction to valorize grape biomass, taking into consideration the aspect of sustainable development together with a circular economy approach.

The first paragraph introduces the reader to the reasons that lead to a necessary radical change in the entire world towards a circular economy approach and greater eco-sustainability of processes. Subsequently, data about production and exploitation of grape biomass together with its main bioactive compounds were shown to highlight the interesting characteristics of this biomass and the importance of recovering such compounds from it.

After a brief presentation of non-conventional extraction techniques, subcritical water extraction (SWE) was described, moving ultimately to the grape biomass SWE, which was presented in a comprehensive and careful manner.

 However, I do have some minor concerns that the authors should consider to improve the manuscript before publication. I would recommend a careful revision and editing of the text of the Abstract and Introduction.

Reply: We thank the careful revision made by the reviewer. The manuscript was carefully revised.

Below are some suggested revisions:

Line 24: “has” instead of “as” (?)

Reply: The typo was corrected.

Line 26: “can be” instead of “is”

Reply: The verb was corrected.

Line 28-29: it may be important to cite also the “Green extraction” principles (see comments for Paragraph 3.1)

Reply: We thank the reviewer for the suggestion. The green extraction principles were also cited.

Line 36: remove “population” (repetition)

Reply: The word was removed.

Line 40: “translated” instead of “translates”

Reply: The word was changed.

Line 47: insert a fullstop or a semicolon after [2,5].

Reply: A semicolon was added.

Line 50-52: 55% is related to the reduction of greenhouse gas emissions

Reply: The text was modified to render the idea clearer.

Line 66: “demanded to explore new …..”

Reply: It was changed as requested.

Line 69-71: “a circular economy would encourage” for example….(to remove repetition of “allow”)

Reply: The word was changed.

Line 72-75: can you explain better this part?

Reply: The sentence was changed: “Circular economy allows for economic growth based on the management of global re-sources, avoiding the production and use of new resources, recycling the existent one, generating in this way less waste, which consequently reduces the negative environ-mental impacts caused by them [6].”

Line 88: perhaps “important” instead of “importantly” (?)

Reply: The sentence was change to “…perhaps the most important reason…

Line 137-139: it may be important to cite the first 5/10 importing and exporting countries in the world.

Reply: This information was added to the manuscript.

Line 146: remove “and” after “thus,”.

Reply: The word was removed.

Paragraph 2.1: lines 187-192 and lines 213-219, these two parts are connected, therefore it would be better to put them together and make a synthesis of them.

Reply: The two paragraphs were joined in one: “Bioactive compounds extracted from grape pomace have shown to possess bio-logical activities, such as anti-aging , the ability to protect cells from UV damage, and the promotions of hair growth [41, 44, 53-56], antioxidant [50, 52, 56, 57], anti-inflammatory [50, 52], antimicrobial [57, 58], antiglycant [59], anticancer and cytotoxic [57], as well as beneficial effects both at the hepatic [60] and cardiovascular level [51]; so, grape pomace produced extracts can exhibit these potential biological effects, making them attractive for applications in the pharmaceutical [50-52, 59-61], food [62-66] and cosmetic [35, 56, 67-69] industries.”

Line 210: remove this part “however,….used.”

Reply: The sentence was removed.

Paragraph 3.1: in this part (line 249, line 256), as when the Green Chemistry was cited in the review, there is a lack of citations regarding the Green extraction concept and principles. Therefore, I would suggest including some references about this very important topic, which has paved the way for all non-conventional extraction techniques. I include some examples below:

-Farid Chemat, Maryline Abert Vian, and Giancarlo Cravotto. Green Extraction of Natural Products: Concept and Principles Int J Mol Sci. 2012; 13(7): 8615–8627.

-Green Extraction of Natural Products: Theory and Practice, First Edition. Edited by Farid Chemat and Jochen Strube. 2015 Wiley-VCH Verlag GmbH & Co. Chapter 1. Green Extraction: From Concepts to Research, Education, and Economical Opportunities, Farid Chemat, Natacha Rombaut, Anne-Sylvie Fabiano-Tixier, Jean T.Pierson, and Antoine Bily

-A review of sustainable and intensified techniques for extraction of food and natural products. Farid Chemat, Maryline Abert Vian, Anne-Sylvie Fabiano-Tixier, Marinela Nutrizio, Anet Režek Jambrak, Paulo E. S. Munekata, Jose M. Lorenzo, Francisco J. Barba, Arianna Binello and Giancarlo Cravotto Green Chem., 2020,22, 2325-2353

Reply: We thank the reviewer for the suggestion. This paragraph was changed, and the references were added to the text.

Line 385: remove repetition “production..products..”

Reply: The sentence was changed.

Line 538: introduce a spacing between 3 and Hz

Reply: the space was introduced.

Line 562: mL instead of Ml

Reply: the typo was corrected.

Paragraph 4:

Lines 632-644 Please, introduce some comments about Green extraction principles too.

Reply: We thank the reviewer for the suggestion. Some comments on Green Extraction principles were added.

Lines 642-647: this part is more suitable for the introduction, not for the final remarks.

Reply: We thank the reviewer for the comment, however in this section we want to discuss some of the improvements that can be done in SWE. In this way we think that the introduction of the possibility of use of different kind of solvents, such as neoteric solvents, is an added value to the discussion.

lines 662-664: this phrase is very similar to the one found in the ref. 131.

Reply: The sentence was changed.

The authors could better underline what the prospects of SWE are, and what are the shortcomings in understanding this extraction technique and applying it in a sustainable process.

Reply: We thank the reviewer for the suggestion. Some comments on subcritical water extraction prospects, limitations and their application in sustainable process were briefly discussed.

Comments on the Quality of English Language

I would suggest a moderate editing of English language, especially for the Abstract and Introduction paragraphs.

Reply: We thank the reviewer suggestion. The manuscript was carefully revised.

Reviewer 3 Report

Comments and Suggestions for Authors

Please add energy consumption of proposed techniques, and carbon footprint of proposed extractions.

What would be benefits of usage of those techniques and are the yields of extraction satisfactory?

Comments on the Quality of English Language

Minor editing.

Author Response

Please add energy consumption of proposed techniques, and carbon footprint of proposed extractions.

Reply: We thank the reviewer the suggestion, however there are still a lack of knowledge about the energy consumption and carbon footprint of this methods. Life cycle assessment can give an idea of this parameters but is very much dependent on the process used, from the raw material to the price of the energy in the country where the experiments are held, to the temperature reached during the experiments. As we discuss in the final section of this review, it is not our goal to give a comprehensive analysis of green chemistry metrics, however we do believe that this area should be more explored if we really want to have green extraction techniques.

What would be benefits of usage of those techniques and are the yields of extraction satisfactory?

Reply: We thank the reviewer comment. The main advantages of SWE use compared to other recent or more traditional extraction techniques were highlighted throughout the sections 3.1 and 3.2, which also discuss the differences between the extraction yields obtained for each tested technique. However, it is our opinion that one of the most important benefits is the widely application of the technique to recover different type of compounds from very different type of matrices mostly using different pressures and temperatures. In addition, the obtained extracts are in water which turns them very easy to apply and incorporate in different type of food, cosmetic, nutraceutical or pharmaceutical products demonstrating its high application potential. Regarding the question if the yields of extraction are satisfactory, most of the research demonstrate that SWE enable to recover higher amounts of bioactive compounds; however, this is mostly connected to the extraction conditions employed and the type of compounds to be recovered which highly influences the final yield of extraction. So, it is very difficult to conclude if the yields are good. Further, it is crucial to evaluate the extraction yields at pilot and larger scales. Most of the studies were performed at lab scale, and extractions at larger conditions will influence the extraction yields.
